# Asymptomatic bacteriuria and its associated fetomaternal outcomes among pregnant women delivering at Bugando Medical Centre in Mwanza, Tanzania

**Colman Mayomba[1], Dismas Matovelo[1]\*, Richard Kiritta[1], Zengo Kashinje[2], Jeremiah Seni[2]**

1 Department of Obstetrics and Gynecology, Weill Bugando School of Medicine, Catholic University of Health and Allied Sciences, Mwanza, Tanzania, 2 Department of Microbiology and Immunology, Weill Bugando School of Medicine, Catholic University of Health and Allied Sciences, Mwanza, Tanzania

\* magonza@bugando.ac.tz

**Data Availability Statement:** All relevant data are within the paper and its Supporting information.

## Abstract

### Background

Asymptomatic bacteriuria (ASB) affects 23.9% of pregnant women globally and, if left untreated, can lead to adverse fetomaternal outcomes. In Tanzania, ASB prevalence has ranged from 13% to 17% over the past decade. Yet, its impact on fetomaternal outcomes remains unexplored, hindering the development of screening strategies, antimicrobial therapies, and preventive measures for this vulnerable population.

### Methods

A cross-sectional analytical study was conducted on 1,093 pregnant women admitted for delivery at Bugando Medical Center (BMC) in Mwanza, Tanzania, from July to December 2022. Socio-demographic, obstetric, and clinical data were collected from the women, along with mid-stream urine samples for analysis. Fetomaternal outcomes were assessed within 72 hours after delivery.

### Results

The median age of participants was 29 years (range: 15–45 years). ASB prevalence among pregnant women was 16.9% (185/1093), with a 95% CI of 14.6–19.3%. Risk factors for ASB included anemia (OR: 5.3; 95% CI = 3.7–8.2, p-value <0.001) and a history of antenatal care admission (OR 4.2; 95% CI = 2.9–6.1, p-value <0.001). Among all participants, 82 (7.5%), 65 (5.9%), 49 (4.5%), and 79 (7.2%) experienced pre-term labor (PTL), premature rupture of membrane (PROM), preeclampsia, and delivered newborns with low birthweight (LBW), respectively. Among the 185 patients with ASB, the respective proportions of PTL, PROM, preeclampsia, and LBW were 25.4%, 17.3%, 9.2%, and 12.4%. Multivariable logistic regression analysis revealed significant associations between ASB and PTL [OR (95% CI): 8.8 (5.5–14.5); p-value <0.001], PROM [OR (95% CI): 4.5 (2.5–8.0); p-value <0.001],

**Funding:** The author(s) received no specific funding for this work.

**Competing interests:** The authors have declared that no either financial or non-financial competing interests exist.

**Abbreviations:** ACOG, American College of Obstetrics and Gynecology; AMR, Antimicrobial resistance; APGAR, Appearance, Pulse, Grimace, Activity, Respiration; ASB, Asymptomatic Bacteriuria; BMC, Bugando Medical Centre; C/S, Caesarean Section; CUHAS, Catholic University of Health and Allied Sciences; Hb, Haemoglobin; IUFD, Intrauterine Fetal Death; IUGR, Intrauterine Growth Restriction; LBW, Low Birth Weight; LMICs, Low-Middle Income Countries; LNMP, Last Normal Menstrual Period; MSU, Mid-Stream Urine; NICU, Neonatal Intensive Care Unit; PE-E, Pre-eclampsia-Eclampsia; PROM, Premature Rupture of Membranes; PTL, Preterm Labour; SB, Symptomatic bacteriuria; UTIs, Urinary Tract Infections; WHO, World Health Organization.

and LBW [OR (95% CI): 2.0 (1.2–3.5); p-value = 0.011]. *Escherichia coli* (50.8%) and *Klebsiella pneumoniae* (14.05%) were the most common pathogens, with low resistance rates to nitrofurantoin, amoxicillin-clavulanate, and cephalosporins—antibiotics considered safe during pregnancy—ranging from 8.2% to 31.0%.

## Conclusion

The prevalence of ASB among pregnant women in Tanzania remains high and is associated with adverse fetomaternal outcomes. Integrating routine urine culture screening for all pregnant women, irrespective of symptoms, and providing specific antimicrobial therapies during antenatal care can help prevent adverse pregnancy outcomes.

## Introduction

Asymptomatic bacteriuria (ASB) affects 23.9% of pregnant women globally and is highly associated with a history of prior urinary tract infection (UTI), preexisting diabetes mellitus (DM), increased parity, and low socioeconomic status [1]. In low- and middle-income countries (LMICs), ASB prevalence during pregnancy has been reported to range broadly from 9% to 86.6%. This prevalence is particularly challenging due to multi-drug resistant pathogens such as *Escherichia coli*, *Staphylococcus spp*, and *Enterococcus* spp, which limit antimicrobial therapeutic options. In Tanzania, ASB prevalence in pregnancy has been reported to range from 9% to 17% [2–4]. Although ASB and subsequent UTI can occur in the general population, they are more prevalent in pregnant women due to various physiological and hormonal changes, impaired immunity, and the physical effects of the gravid uterus on the urinary bladder and ureter [1, 5, 6].

During pregnancy, specific physiological changes occur, including increased serum progesterone levels leading to relaxation of urinary tract smooth muscles, elevated urine alkalinity due to increased excretion of bicarbonates, and mechanical compression of the urinary bladder and ureters by the gravid uterus. These changes create favorable conditions for pathogens to colonize the urinary bladder, multiply, trigger an inflammatory response, and subsequently lead to UTIs [5]. The asymptomatic clinical state of ASB has been attributed to the absence of Type 1 fimbriae in certain strains of bacteria, particularly *E. coli*. This absence results in a lack of immunological response, contributing to the absence of symptoms [7]. If left untreated, ASB in pregnancy can lead to various adverse effects, including pyelonephritis, preeclampsia (PE), anemia, low birth weight (LBW), intrauterine growth restriction (IUGR), preterm labor (PTL), premature rupture of membranes (PPROM), endometritis, as well as puerperal and neonatal sepsis [8].

Despite the high burden of ASB in Tanzania, the national guideline for antenatal care (ANC) recommends urine dipstick and microscopic urinalysis for UTI screening in pregnancy. However, both tests have been reported to have low sensitivity for detecting ASB in pregnancy. Despite extensive studies on urinary tract infections (UTIs) in various populations, including pregnant women [2–4, 9, 10], the impact of UTIs on pregnancy outcomes has not been evaluated in Tanzania. To address this critical research gap, our study was designed to determine the prevalence of ASB among pregnant women admitted for delivery at Bugando Medical Centre (BMC), assess its associated adverse fetomaternal outcomes, and analyze the antimicrobial susceptibility profiles of implicated pathogens. These findings aim to guide evidence-based screening and management strategies for UTIs in pregnant women.

## Methods

### Study design

We conducted a cross-sectional analytical study from July 5th, 2022, to December 15th, 2022, at the labor and delivery wards of Bugando Medical Centre (BMC).

### Study setting

BMC serves as a tertiary zonal referral and teaching hospital for the Catholic University of Health and Allied Sciences (CUHAS). BMC serves the northwestern region of Tanzania with a catchment area of approximately 20 million people across eight administrative regions. BMC has a total bed capacity of 1,000 and conducts approximately 500 deliveries per month (source: https://bmc.go.tz/public/). During the enrollment period, all eligible patients were provided with detailed information about the study procedures and were invited to participate. Those who voluntarily agreed to participate signed the informed consent form.

### Study participants

Inclusion criteria comprised pregnant women with a gestational age >28 weeks who were admitted for delivery and were in the latent phase of labor or planned for delivery at BMC.

### Eligibility criteria

These women should not have exhibited any symptoms suggestive of urinary tract infection (UTI) such as dysuria, pyuria, or haematuria, with or without fever. Exclusion criteria included pregnant women with multiple pregnancies, HIV infection, sickle cell anemia, and diabetes mellitus in the index pregnancy.

### Sample size calculation and sampling technique

The sample size was determined using the Leslie Kish formula with an expected prevalence of ASB of 17% [2] and a standard error of 0.05, resulting in a minimum sample size of 216. For extrapolating adverse outcomes, the sample size was calculated using the formula for comparing two proportions [11]: $n = (Z\alpha/2 + Z\beta)^2 * (p1(1-p1) + p2(1-p2)) / (p1-p2)$, where $Z\alpha/2$ represents the critical value of the normal distribution at $\alpha/2$ (for a confidence level of 95%, $\alpha = 0.05$, and $Z\alpha/2 = 1.96$), and $Z\beta$ represents the critical value of the normal distribution at $\beta$ (for a power of 80%, $\beta = 0.2$, and $Z\beta = 0.84$). P1 was set at 0.133 and P2 was set at 0.0076, representing the proportion of women with asymptomatic bacteriuria delivering newborns with low birthweight and the proportion of women without asymptomatic bacteriuria delivering newborns with low birthweight, respectively, based on a similar study conducted in Israel. Taking into account the prevalence of preterm labor of 11.7% in Kilimanjaro, Tanzania [12] as one of the adverse pregnancy outcomes. The projected minimum sample sizes in the two groups required to observe a significant difference at a power of 80% and a 95% confidence level were determined to be 83 and 626, respectively. A convenience sampling method was employed to recruit eligible patients until the desired sample size of 1103 was achieved.

### Data collection procedures

Out of 1,103 participants who met the inclusion criteria, 10 were excluded due to incomplete medical information, leaving 1,093 participants who were enrolled and included in this study. Socio-demographic, obstetric data, and other clinical information were collected using pretested structured questionnaires after obtaining voluntary informed consent. Urine sample

collection followed CUHAS Urine sample protocol, (supporting file: S1 File urine culture protocol) in which pregnant women were instructed to clean the genital area with water from front to back, part the labia, and collect clean catch mid-stream urine in a sterile screw-capped transparent container. Urine samples were transported to the laboratory within 2 hours of collection in a cold box. Conventional Kass quantitative urine culture was performed for each sample using standard guidelines on Blood agar and MacConkey agar (OXOID, UK), and incubated at 35 to 37˚C for 18 to 24 hours. ASB was diagnosed when there was significant bacteriuria of $10^5$ colony-forming units (CFU) per milliliter volume of urine of one or two bacterial species. Biochemical identification tests were conducted for Gram-positive and Gram-negative bacteria using standard methods [2, 13]. Antimicrobial susceptibility tests were performed on all identified bacteria using the conventional Kirby-Bauer disk diffusion method on Muller Hinton Agar (OXOID, UK). The respective zones of inhibition or diameters were interpreted as resistance, intermediate, or sensitive using the Clinical Laboratory Standard Institute (CLSI) guidelines for specific management [14, 15].

Fetomaternal outcomes among participants were recorded within 72 hours after delivery, and information was abstracted from patients' medical records based on attending obstetricians' diagnoses. Laboratory reports on isolated bacteria and antimicrobial sensitivity patterns were promptly submitted to the attending doctors to guide specific management. For participants who experienced false labor and opted for discharge, their antenatal care (ANC) cards were labeled, and the investigator's phone number was documented on the ANC card for tracking purposes in the event of readmission.

## Data management and analysis

All socio-demographic, obstetric, and other clinical data collected were entered into Microsoft Excel® for consistency checks and then analyzed using STATA software version 15®. Descriptive analyses were conducted using means (± standard deviation), frequencies, and proportions. The ASB and adverse fetomaternal outcomes (preterm delivery, premature rupture of membranes, preeclampsia, and low birthweight) were analyzed as outcome variables using bivariable logistic regression analysis against socio-demographic, obstetric, and clinical characteristics as potential predictors. All independent variables with a p-value < 0.2 were subjected to multivariable logistic regression analysis. Independent predictors of adverse fetomaternal outcomes were determined using odds ratios and 95% confidence intervals, with a cut-off p-value of $\leq 0.05$.

## Ethical considerations

This study received approval from the joint CUHAS/BMC Research Ethics and Review Committee (CREC/561/2022). Permission to conduct the study was obtained from the BMC Director General and the Head of the Department of Obstetrics and Gynecology before the commencement of the study. Written informed consent was obtained from willing participants aged 18 years old and older. For participants under 18 years of age, voluntary assent was obtained, and informed consent was provided by their respective parents or guardians. Results from culture and antimicrobial susceptibility testing were promptly provided to attending doctors to guide patient management.

## Results

### Socio-demographic and clinical characteristics of enrolled pregnant women

The mean age of the participants was 29.0 ± 5.3 years, ranging from 15 to 45 years. The majority of women resided in urban areas (92.4%) and were married (91.6%). Approximately two-

**Table 1. Socio-demographic and clinical characteristics of study participants (N = 1,093).**

| Characteristics | | Frequency (N) | Percentage (%) |
|---|---|---|---|
| **Social demographics** | | | |
| Age(Years) | <30 | 534 | 48.0 |
| | ≥30 | 579 | 52.0 |
| Residency | Urban | 1010 | 92.4 |
| | Rural | 83 | 7.6 |
| Education Level | Primary | 172 | 15.7 |
| | Secondary | 762 | 69.8 |
| | Tertiary | 159 | 14.6 |
| Occupation | Housewife | 76 | 7.0 |
| | Self-employed | 880 | 81.0 |
| | Employed | 137 | 12.0 |
| Marital status | Married | 1000 | 91.6 |
| | Not married* | 93 | 8.4 |
| **Clinical characteristics of study participants** | | | |
| GA(Weeks) | ≥37 | 1011 | 92.5 |
| | <37 | 82 | 7.5 |
| Gravidity | Multigravida | 753 | 68.9 |
| | Primigravida | 350 | 31.1 |
| History-of admission | Admitted | 200 | 18.3 |
| | Not Admitted | 893 | 81.7 |
| Anemia-in pregnancy | Anemic | 146 | 13.4 |
| | Not anemic | 947 | 86.5 |
| Malaria-in pregnancy | Positive | 1 | 0.1 |
| | Negative | 1092 | 99.9 |
| History of antibiotic use | Used | 149 | 13.6 |
| | Not used | 944 | 86.4 |
| ASB | Positive | 185 | 16.9 |
| | Negative | 908 | 83.1 |

*Means combined groups of single, cohabiting, divorced, and widow

thirds were multigravida (68.9%), 13.4% had anemia, and 18.3% had a history of hospital admission in the past three months (Table 1).

## Prevalence and risk factors of asymptomatic bacteriuria

The overall prevalence of ASB was 16.9% (185/1093), with a 95% confidence interval of 14.6–19.3%. Multivariable logistic analysis revealed that having anemia during antenatal care (ANC) visit [OR: 5.3; 95% CI = 3.7–8.2, p-value <0.001] and history of admission [OR 4.2; 95% CI = 2.9–6.1, p-value <0.001] were significantly associated with ASB (Table 2).

## Adverse maternal outcomes among pregnant women with asymptomatic bacteriuria at BMC

A total of 82 (7.51%), 65 (5.95%), and 49 (4.49%) participants experienced preterm labor (PTL), premature rupture of membranes (PROM), and preeclampsia, respectively. Among the 185 participants with ASB, 47 (25.41%) had PTL, 32 (17.3%) had PROM, and 17 (9.19%) had

**Table 2. Multivariable logistic regression analysis on predictor variables for asymptomatic bacteriuria.**

| Variable | Asymptomatic bacteriuria | | Bivariable | | Multivariable | |
|---|---|---|---|---|---|---|
| | Negative | Positive | OR [95%CI] | p-value | OR [95%CI] | p-value |
| | n (%) | n (%) | | | | |
| **Age** | | | | | | |
| <30 | 447 (82.9) | 92 (17.1) | 1 | | | |
| ≥30 | 461 (83.2) | 93 (16.8) | 0.90 [0.7–1.3] | 0.912 | | |
| **Residency** | | | | | | |
| Urban | 838 (82.9) | 172 (17.1) | 1 | | | |
| Rural | 70 (84.3) | 13 (15. 7) | 1.10 [0.5–2.4] | 0.751 | | |
| **Education** | | | | | | |
| Primary | 144 (83.7) | 28 (16.3) | 1 | | | |
| Secondary | 625 (82.0) | 137 (18.5) | 1.1 [0.7–1.7] | 0.620 | | |
| Tertiary | 139 (87.4) | 20 (12.6) | 0.7 [0.4–1.4] | 0.341 | | |
| **Occupation** | | | | | | |
| Housewife | 65 (85.5) | 11 (14.5) | 1 | | | |
| Self-employed | 722 (81.7) | 165 (18.3) | 1.3 [0.6–2.6] | 0.407 | | |
| Employed | 118 (90.8) | 12 (9.2) | 0.6 [0.3–1.4] | 0.253 | | |
| **Gravidity** | | | | | | |
| Primigravida | 285 (83.8) | 55 (16.2) | 1 | | | |
| Multigravida | 623 (82.7) | 130 (17.3) | 1.08 [0.7–1.5] | 0.650 | | |
| **History-of admission** | | | | | | |
| No | 783 (87.7) | 110 (12.3) | 1 | | 1 | |
| Yes | 125 (62.5) | 75 (37.5) | 4.3 [3.1–6.1] | <0.001 | 4.2 [2.9–6.1] | <0.001 |
| **Anaemia-in pregnancy** | | | | | | |
| No | 827 (87.3) | 120 (12.7) | 1 | | 1 | |
| Yes | 81 (55.5) | 65 (44.5) | 5 [3.8–8.0] | <0.001 | 5.3 [3.7–8.2] | <0.001 |
| **History-of antibiotics use** | | | | | | |
| Yes | 686 (83.3) | 137 (16.7) | 1 | | | |
| No | 222 (82.2) | 48 (17.8) | 1.08 [0.7–1.6] | 0.671 | | |

preeclampsia. Multivariable logistic regression analysis revealed that ASB was significantly associated with premature rupture of membranes [OR (95% CI): 4.5 (2.5–8.0); p-value <0.001] and preterm labor [OR (95% CI): 8.8 (5.5–14.5); p-value <0.001] (Tables 3 & 4). Additionally, there was no association between various socio-demographic and clinical characteristics of pregnant women with preeclampsia on bivariable logistic regression analysis, except ASB, 9.2% versus 3.2% [OR (95% CI): 2.8 (1.5–5.0); p-value <0.001].

## Adverse fetal outcomes among women with asymptomatic bacteriuria delivering at BMC

The overall rates of stillbirth, birth asphyxia, NICU admission, and early neonatal death were 2.0% (22/1093), 4.6% (50/1093), 8.4% (92/1093), and 4.0% (44/1093), respectively. However, none of these outcomes showed a significant association with ASB among delivering women. Among the participants, 79 (7.23%) women delivered babies with low birth weight (LBW), with a notable increase in LBW observed in babies delivered by women with ASB (12.43% versus 6.2%) [OR (95% CI): 2.0 (1.2–3.5); p-value = 0.011] (Table 5).

**Table 3. Multivariable logistic regression analysis for predictor variables of premature rupture of membrane.**

| Variable | Premature rupture of membrane, n (%) | | Bivariable | | Multivariable | |
|---|---|---|---|---|---|---|
| | No | Yes | OR [95%CI] | p-value | OR [95%CI] | p-value |
| **Age** | | | | | | |
| <30 | 499 (92.6) | 40 (7.4) | 1 | 0.441 | | |
| ≥30 | 529 (92.5) | 25 (7.5) | 0.6 [0.3–1] | | | |
| **Residence** | | | | | | |
| Rural | 80 (96.4) | 3 (3.6) | 1 | 0.363 | | |
| Urban | 948 (93.1) | 65 (6.1) | 1.7 [0.5–5.7] | | | |
| **Education** | | | | | | |
| Primary | 163 (94.8) | 9 (5.2) | 1 | | | |
| Secondary | 712 (93.4) | 50 (6.6) | 1.2 [0.6–2.6] | 0.516 | | |
| Tertiary | 153 (96.2) | 5 (3.8) | 0.7 [0.2–2.0] | 0.526 | | |
| **Occupation** | | | | | | |
| Housewife | 69 (90.8) | 7 (9.2) | 1 | | | |
| Self-employed | 825 (93.8) | 55 (6.2) | 0.6 [0.3–1.5] | 0.318 | | |
| Employed | 128 (97.7) | 3 (2.3) | 0.2 [0.05–0.9] | 0.039 | | |
| **Gravidity** | | | | | | |
| Primigravida | 315 (92.7) | 25 (7.3) | 1 | | | |
| Multigravida | 713 (94.7) | 40 (5.3) | 0.7 [0.4–1.2] | 0.280 | | |
| **History-of admission** | | | | | | |
| No | 851 (95.3) | 42 (4.7) | 1 | | 1 | |
| Yes | 177 (88.5) | 23 (11.5) | 2.6 [1.5–4.5] | 0.001 | 1.7 [0.9–2.9] | 0.081 |
| **Anaemia-in pregnancy** | | | | | | |
| No | 898 (94.8) | 49 (5.2) | 1 | | 1 | |
| Yes | 130 (89.1) | 16 (10.9) | 2.3 [1.2–4] | 0.007 | 1.2 [0.7–2.4] | 0.499 |
| **History-of antibiotics use** | | | | | | |
| Yes | 774 (94.1) | 49 (5.9) | 1 | | | |
| No | 254 (94.1) | 16 (5.9) | 1 [0.6–1.8] | 0.987 | | |
| **Asymptomatic bacteriuria** | | | | | | |
| Negative | 875 (96.4) | 33 (3.6) | 1 | | 1 | |
| Positive | 153 (82.7) | 32(17.3) | 5.5 [5.3–9.3] | <0.001 | 4.5[2.5–8] | 0.001 |

## Antimicrobial resistance patterns of bacteria implicated in asymptomatic bacteriuria

The most commonly isolated bacteria from delivering women with ASB were *Escherichia coli* (50.8%) followed by *Klebsiella pneumoniae* (14.05%). *Enterococcus spp*, *Acinetobacter spp*, and *Staphylococcus aureus* each accounted for 5.41%. High resistance rates were observed for ampicillin/piperacillin and trimethoprim-sulfamethoxazole among all Gram-negative and Gram-positive bacteria. However, notably low resistance rates were observed in *Escherichia coli* and *Klebsiella pneumoniae* for nitrofurantoin (8.2% and 30.1%, respectively), ceftriaxone (13.0% and 19.2%, respectively), gentamicin (5.2% and 7.0%, respectively), piperacillin-tazobactam (3.9% and 15.0%, respectively), and meropenem (0.0% and 10.3%, respectively). Resistance of *Staphylococcus aureus* to cefoxitin was observed in 90% of cases (9/10), indicating Methicillin-resistant *Staphylococcus aureus* (MRSA) isolates. However, resistance to nitrofurantoin among Staphylococcus aureus isolates was low (10.2% and 0.0%, respectively) (Table 6).

**Table 4. Multivariable logistic regression analysis for predictors of preterm labor.**

| Variable | Preterm labor n (%) | | Bivariable | | Multivariable | |
|---|---|---|---|---|---|---|
| | No | Yes | OR [95%CI] | p-value | OR [95%CI] | p-value |
| **Age** | | | | | | |
| <30 | 498 (92.6) | 40 (7.4) | 1 | | | |
| ≥30 | 512 (92.5) | 42 (7.5) | 1 [0.7–1.6] | 0.930 | | |
| **Residence** | | | | | | |
| Rural | 74 (89.2) | 9 (10.8) | 1 | | | |
| Urban | 936 (92.7) | 73 (7.3) | 0.8 [0.3–1.4] | 0.241 | | |
| **Education** | | | | | | |
| Primary | 153 (88.9) | 19 (11.1) | 1 | | 1 | |
| Secondary | 706 (92.9) | 54 (7.1) | 0.6 [0.4–1.1] | 0.085 | 0.5 [0.3–1.0] | 0.051 |
| Tertiary | 151 (94.9) | 8 (5.1) | 0.4 [0.2–1.0] | 0.051 | 0.5 [0.2–1.1] | 0.073 |
| **Occupation** | | | | | | |
| Housewife | 75 (94.7) | 8 (5.3) | 1 | | | |
| Self-employed | 808 (91.9) | 71 (8.1) | 1.6 [0.6–4.5] | 0.386 | | |
| Employed | 125 (95.4) | 6 (4.6) | 0.9 [0.2–3.2] | 0.835 | | |
| **Gravidity** | | | | | | |
| Primigravida | 315 (92.6) | 25 (7.4) | 1 | | | |
| Multigravida | 695 (92.5) | 57 (7.5) | 1.03 [0.6–1.7] | 0.891 | | |
| **History-of admission** | | | | | | |
| No | 833 (93.4) | 59 (6.6) | 1 | | 1 | |
| Yes | 177 (88.5) | 23 (11.5) | 1.8 [1.1–3.1] | 0.190 | 0.9 [0.5–1.6] | 0.836 |
| **Anaemia-in pregnancy** | | | | | | |
| No | 875 (92.5) | 71 (7.5) | 1 | | | |
| Yes | 135 (92.5) | 11 (7.5) | 1 [0.5–1.9] | 0.990 | | |
| **History-of antibiotics use** | | | | | | |
| No | 754 (91.7) | 68 (8.3) | 1 | | 1 | |
| Yes | 256 (94.8) | 14 (5.2) | 0.8 [0.3–1.1] | 0.098 | 0.6 [0.3–1.02] | 0.061 |
| **Asymptomatic bacteriuria** | | | | | | |
| Negative | 873 (96.2) | 35 (3.9) | 1 | | 1 | |
| Positive | 138 (74.6) | 47 (25.4) | 8.5 [5.3–13.6] | <0.001 | 8.8 [5.5–14.5] | <0.001 |

## Discussion

### The burden of asymptomatic bacteriuria among pregnant women

The study enrolled pregnant women without specific signs and symptoms of urinary tract infection (UTI) in pregnancy, and, consistent with other studies in Sub-Saharan Africa [16, 17], found a prevalence of ASB of approximately 17%. This prevalence is comparable to previous studies conducted in the same hospital, where the burden of ASB and symptomatic bacteriuria was reported as 13% and 17.9% [5], and 17.7% and 17.6% five years ago [2], respectively. The persistent high prevalence of ASB is likely due to a lack of routine screening and treatment of ASB in pregnancy at BMC and across Tanzania. Although the Tanzania ANC guideline recommends microscopic urinalysis and urine dipstick for screening ASB [18], these tests have been reported in several studies to have poor sensitivity, with sensitivities of 25% and 38% [4], respectively. Therefore, our findings emphasize the need to incorporate urine culture into the ANC package for ASB screening, as it is the gold standard test with a sensitivity of >96% [19].

**Table 5. Multivariable logistic regression analysis for predictor variables of low birth weight.**

| Variable | Low birth weight N (%) | | Bivariable | | Multivariable | |
|---|---|---|---|---|---|---|
| | No | Yes | OR [95% CI] | p-value | OR[95% CI] | p-value |
| **Age** | | | | | | |
| <30 | 502 (93.1) | 37 (6.9) | 1 | | | |
| ≥30 | 512 (92.4) | 42 (7.6) | 1.1 [0.7–1.8] | 0.651 | | |
| **Residence** | | | | | | |
| Rural | 75 (90.4) | 8 (9.6) | 1 | | | |
| Urban | 939 (92.9) | 71 (7.1) | 0.7 [0.3–1.5] | 0.380 | | |
| **Education** | | | | | | |
| Primary | 161 (93.6) | 11 (6.4) | 1 | | | |
| Secondary | 707 (92.8) | 55 (7.2) | 1.2 [0.6–2.2] | 0.701 | | |
| Tertiary | 146 (91.8) | 13 (8.2) | 1.3 [0.6–2.9] | 0.533 | | |
| **Occupation** | | | | | | |
| Housewife | 72 (94.7) | 4 (5.3) | 1 | | | |
| Self-employed | 816 (92.7) | 64 (7.3) | 1.4 [0.5–4] | 0.515 | | |
| Employed | 120 (91.6) | 11 (8.4) | 1.7 [0.5–5.4] | 0.398 | | |
| **Gravidity** | | | | | | |
| Primigravida | 5 (95.7) | 23 (4.3) | 1 | | | |
| Multigravida | 528 (95.3) | 26 (4.7) | 1.1[0.6–1.9] | 0.250 | | |
| **History-of admission** | | | | | | |
| No | 834 (93.4) | 59 (6.6) | 1 | | 1 | |
| Yes | 180 (90) | 20 (10) | 1.5 [0.9–2.7] | 0.096 | 1.2[0.7–2.5] | 0.374 |
| **Anaemia-in pregnancy** | | | | | | |
| No | 879 (92.8) | 68 (7.2) | 1 | | | |
| Yes | 125 (92.5) | 11 (7.5) | 1.0 [0.5–2] | 0.878 | | |
| **History-of antibiotics use** | | | | | | |
| Yes | 784 (95.8) | 38 (4.2) | 1 | | | |
| No | 250 (94.4) | 20 (5.6) | 1.3 [0.7–2.5] | 0.250 | | |
| **ASB** | | | | | | |
| Negative | 852 (93.8) | 56 (6.17) | 1 | | 1 | |
| Positive | 162 (87.5) | 23 (12.4) | 2.2 [1.3–3.6] | 0.003 | 2.0 [1.2–3.5] | 0.011 |

The magnitude of ASB prevalence in this study is higher than that reported in other recent studies. For instance, a study conducted in 2018 at the Kilimanjaro Christian Medical Centre (KCMC) in Tanzania reported a prevalence of 8.5% [3], while another in Mbale, Uganda reported a prevalence of 3.7% [20]. The elevated prevalence we found may be explained by the fact that our study enrolled participants in the third trimester, while the two studies enrolled participants in the first and second trimester only. It is known that as pregnancy advances, more hormones are released by the growing placenta, leading to the relaxation of smooth muscles in the urinary tract and urine stasis. These factors can contribute to increased bacterial multiplication [5]. Urine stasis is also caused by poor perineal hygiene and physical compression of the ureters and urinary bladder by the enlarged gravid uterus. These factors, along with poor perineal hygiene, can lead to increased bacterial multiplication, inflammatory responses, and ultimately urinary tract infections (UTIs) [5, 21]. An increase in urine culture positivity is also evident in the third trimester, potentially explaining the higher prevalence of ASB observed in our study [22]. The prevalence in this study is also higher than the prevalence reported in the USA and Canada [23, 24]. These differences may be accounted for by varying epidemiology of the disease across countries; in developed countries, screening, and treatment of ASB have been

**Table 6. Antimicrobial resistance patterns of bacteria implicated in asymptomatic bacteriuria.**

| Bacteria species (n)% | Antimicrobial resistance patterns (%) | | | | | | | | | | |
|---|---|---|---|---|---|---|---|---|---|---|---|
| | NIT | AMP/PL | AMC | CRO/CAZ¥ | CIP | TZP | MEM | GEN | ERY | VA | SXT |
| *Escherichia coli*, (94) 50.8% | 8.2 | 73.6 | 15.3 | 13.0 | 20.3 | 3.9 | 0.0 | 5.2 | NA | NA | 72.3 |
| *Klebsiella pneumoniae*, (26) 14.1% | 30.1 | NA | 31.0 | 19.2 | 14.8 | 15.0 | 10.3 | 7.0 | NA | NA | 57.8 |
| *Acinetobacter spp*, (10) 5.41% | NA | 90.1* | NA | 60.2 | 20.3 | 10.1 | 20.4 | 19.7 | NA | NA | 39.9 |
| *Enterococcus spp*, (10) 5.41% | 40.1 | 69.8 | NA | NA | 19.7 | NA | NA | 0.0** | 20.1 | 30.2 | 70.0 |
| *Staphylococcus aureus*, (10) 5.4% | 10.2 | 90.3 | NA | NA | 9.9 | NA | NA | 0.0 | 80.0 | NT | 60.0 |
| Other GNB, (28) 15.14% | 27.3 | 85.3* | 25.82 | 22.2 | 21.3 | 5.2 | 15.1 | 21.1 | NA | NA | 68.0 |
| Other GPB, (7) 3.78% | 20.1 | 20.1 | NA | NA | 0.0 | NA | NA | 40.0 | 33.3 | NA | 85.7 |

NIT: nitrofurantoin, AMP: ampicillin, PL: piperacillin, AMC: amoxicillin-clavulanic acid, CRO: ceftriaxone, CIP: ciprofloxacin, TZP: piperacillin-tazobactam, MEM: meropenem, GEN: gentamicin, SXT: trimethoprim-sulfamethoxazole, VA: vancomycin, SXT: Trimethoprim-sulphamethoxazole, R: resistance, NA: Not Applicable, NT: Not tested, GNB: negative bacteria [*Enterobacter spp*. (3), *Proteus mirabilis* (1), *Pseudomonas aeruginosa* (2), *Klebsiella oxytoca* (2), *Aeromonas hydrophilia* (1) *and unidentified GNB* (19)]; GPB: Gram-positive bacteria: [*Streptococcus spp* (5) *Streptococcus pyogenes* (2)];

\**Piperacillin has used for Acinetobacter spp*. and *Pseudomonas aeruginosa* isolates;

\** High-level Gentamicin (120μg) for Enterococcus spp was used;

¥*CAZ: 3^rd generation cephalosporin ceftazidime was used for Acinetobacter spp*. and *Pseudomonas aeruginosa isolates*. All results can be found in the supporting file labelled S2 File: ASB Study database.

incorporated into ANC guidelines, leading to a lower burden of ASB in pregnant women. Additionally, it is reported that ASB is significantly associated with poor socioeconomic status, contributing to the preponderance of ASB among pregnant women in developing countries [22].

## Risk factors associated with significant asymptomatic bacteriuria

Having anemia during antenatal care (ANC) visits and a history of prior hospital admissions were found to be significantly associated with developing ASB, a linkage supported by several studies. Bacterial endotoxins produced by Gram-negative bacteria persistently damage red cell membranes, leading to early cell death and subsequent anemia [21]. Inflammatory responses due to ASB are associated with the release of IL-1, which directly lowers erythropoietin secretion. Additionally, IL-1 and TNF-α, through interferon-γ, can lead to the suppression of bone marrow response to erythropoietin. Furthermore, persistent bacterial infection also induces the liver to produce the protein hepcidin, which inhibits iron absorption from the gut and releases iron from its storage sites, resulting in anemia in pregnant women with ASB [21]. However, this study was cross-sectional in design, assessing both exposure and outcome simultaneously. This design limitation restricts our ability to fully explain the interplay between anemia and ASB and to establish any potential causal relationship. There was a significant association between ASB and a history of admission, as documented in a previous study in the same region [2]. This association may be attributed to invasive procedures conducted during hospitalization, such as intravenous cannulation and urethral catheterization, as well as the limited hygiene and sanitation practices that may be present in a congested hospital environment. Together, these factors may predispose recently admitted pregnant women to ASB.

## Adverse fetomaternal outcomes among women with asymptomatic bacteriuria

In this study, PTL, PROM, and pre-eclampsia were significantly associated with ASB, highlighting the importance of introducing screening and treatment strategies for ASB in

pregnancy as part of the core antenatal care (ANC) package in Mwanza, Tanzania, and other regions with similar epidemiological predispositions. Similar findings were observed in two studies conducted in India [22], and another study in Israel. In a meta-analysis, the chances of preterm delivery and low birth weight infants were one-half and two-thirds lower among women without bacteriuria compared to those with significant bacteriuria (SB) [17]. In this study, patients with ASB had three times the odds of having preeclampsia on bivariate analysis, consistent with a 1:2 matched case-control study conducted in the same hospital in 2017. In that study, 50.4% of cases with preeclampsia had SB compared to 16.8% in the control group, with 7.7 odds of having bacteriuria among pregnant women with preeclampsia compared to those without preeclampsia [25].

The mechanism by which ASB causes PTL is unknown, although it is thought to be due to the release of proinflammatory cytokines by maternal and fetal monocytes in reaction to bacterial products such as phospholipase A2. This results in elevated prostaglandins within the uterus and, therefore, PTL. Furthermore, ASB triggers PROM by the release of bacterial endotoxins, proteases, and collagenases, which directly degrade and digest fetal membranes, resulting in PROM [26]. Additionally, ASB causes chronic subclinical infection, which in turn leads to increased maternal cytokine levels sufficient to affect vascular endothelial function and, hence, preeclampsia.

Low birth weight (LBW) was significantly associated with ASB in this study. Of all patients enrolled, 79 (7.23%) had LBW, and 23 (12.43%) babies delivered by pregnant women with ASB had LBW. These findings are comparable to studies conducted in India, where the risk of LBW was found to be significant in women with ASB (20.0%) compared to the negative group (8.0%) [22]. Additionally, it has been reported that both ASB and PROM can lead to preterm labor, resulting in the delivery of babies with LBW [27]. Ascending infections in the decidual, chorion, and fetus can lead to the release of bacterial endotoxins. These endotoxins initiate a cascade of inflammatory mediators, leading to circulatory disturbances in the placenta, causing placental insufficiency, and ultimately resulting in LBW [28].

## Antimicrobial resistance patterns of bacteria implicated in asymptomatic bacteriuria

The most common bacteria isolated from the urine were *Escherichia coli* and *Klebsiella pneumoniae*, consistent with findings from previous studies conducted in Tanzania [2, 3]. These findings align with those of a systematic review of Africa [17]. The high frequency of Escherichia coli and Klebsiella pneumoniae, common gut microbiota, may be attributed to limited hygienic and sanitation practices among pregnant women. Additionally, the anatomical proximity between the perineal region and the urogenital opening increases the likelihood of contamination and subsequent ascending infections [3].

Antimicrobial resistance (AMR) was lowest for the predominant Gram-negative bacteria against meropenem, gentamicin, piperacillin-tazobactam, third-generation cephalosporin, amoxicillin-clavulanate, and nitrofurantoin, while resistance was highest for ampicillin and trimethoprim-sulfamethoxazole. This finding is consistent with a previous study conducted at BMC. Notably, the resistance of *Escherichia coli* against nitrofurantoin has slightly decreased from 12.8% to 8.2% [2]. The resistance trends for *Escherichia coli* against antibiotics observed in the current study are consistent with those reported in a previous multicenter study conducted in the same region, spanning from tertiary to primary healthcare facilities [2].

The resistance of *Escherichia coli* and *Klebsiella pneumoniae* against commonly used third generation cephalosporins, notably ceftriaxone, was 13.0% and 19.2%, respectively. These findings align with the recommendations of the Tanzania Standard Treatment Guidelines

and Essential Medicines List for Tanzania Mainland (STG/NEMLIT, 2021), which advocate for nitrofurantoin and amoxicillin-clavulanate as first-line treatments for UTIs, with gentamicin and third generation cephalosporins as second-line options. Meropenem is reserved as a third-line treatment for severe and critically invasive infections such as urosepsis, typically seen in ICU patients. In addition, we report low resistance to piperacillin-tazobactam (3.9% to 15.0%).

The high resistance of *Acinetobacter* spp to various agents is expected, given its preponderance for multidrug resistance, highlighting the need for laboratory-guided antimicrobial therapies at BMC. Conversely, nitrofurantoin and gentamicin demonstrated good sensitivity against Gram-positive bacteria such as *Staphylococcus aureus* and *Enterococcus* spp, making them suitable options for managing patients with ASB caused by these pathogens.

The observed low resistance of *Escherichia coli* and *Klebsiella pneumoniae* to third generation cephalosporins (0.0% to 29.0%) and meropenem (0.0% to 4.8%) is consistent with findings from studies conducted in Ethiopia and Kenya. However, the Ethiopia study reported relatively high resistance to gentamicin (52.0% to 71.0%), and another study in Ghana found high resistance to gentamicin and third-generation cephalosporins [29]. Regional variations in resistance profiles may be attributed to differences in the number of study participants, which impacts the number of bacteria isolates recovered. This underscores the critical need for a large sample size, as demonstrated in the current study, to provide robust data for informing country-specific treatment guidelines.

## Study limitations

The study was designed to assess exposure and outcome concurrently, limiting the capture of long-term adverse pregnancy outcomes. Future research could benefit from a long-term prospective cohort study to address this limitation. Additionally, hemoglobin levels were recorded from ANC cards, which may be subject to inter-trimester and inter-machine variabilities. While efforts were made to minimize these variations by using the most recent hemoglobin measurement, some degree of variability may still exist.

## Conclusion

This study revealed a high prevalence of ASB in pregnancy at a tertiary-level health center in Northwestern Tanzania. History of prior admission and anemia during pregnancy were identified as associated factors with ASB. Importantly, for the first time in Tanzania, this study demonstrated significant associations between ASB and adverse fetomaternal outcomes, including preterm labor (PTL), premature rupture of membranes (PROM), preeclampsia, and low birth weight (LBW).

Given that antimicrobial therapies can potentially mitigate the sequelae of ASB in pregnancy, this study recommends routine screening and treatment of ASB as an integral component of antenatal care (ANC) in Tanzania. In resource-constrained settings, prioritizing ASB screening for patients with identified risk factors such as anemia during pregnancy and previous hospital admissions is advisable.

Based on the findings, nitrofurantoin is suggested as a first-line drug, with third generation cephalosporins (e.g., ceftriaxone) or gentamicin as second-line options, and piperacillin-tazobactam and meropenem as a third-line choice. Furthermore, a prospective cohort study is recommended to evaluate long-term adverse fetomaternal outcomes, which could inform practices at lower-tier healthcare facilities.

## Supporting information

**S1 File.**
(PDF)

**S2 File.**
(XLSX)

## Acknowledgments

The authors would like to acknowledge the support provided by the members of Obstetrics & Gynecology, and Microbiology & Immunology Departments at BMC and CUHAS, and all study participants. Furthermore, we sincerely convey our sincere gratitude to Quinn Goddard from the University of Calgary, Canada, and Megan Willkens from Weill Cornell Medical University, New York for the grammar and scientific writing review.

## Author Contributions

**Conceptualization:** Colman Mayomba, Dismas Matovelo, Jeremiah Seni.

**Data curation:** Colman Mayomba, Dismas Matovelo, Richard Kiritta, Zengo Kashinje, Jeremiah Seni.

**Formal analysis:** Colman Mayomba, Dismas Matovelo, Jeremiah Seni.

**Funding acquisition:** Colman Mayomba, Dismas Matovelo, Jeremiah Seni.

**Investigation:** Colman Mayomba, Dismas Matovelo, Richard Kiritta, Zengo Kashinje, Jeremiah Seni.

**Methodology:** Dismas Matovelo, Jeremiah Seni.

**Project administration:** Dismas Matovelo, Jeremiah Seni.

**Resources:** Zengo Kashinje, Jeremiah Seni.

**Software:** Jeremiah Seni.

**Supervision:** Dismas Matovelo, Jeremiah Seni.

**Validation:** Richard Kiritta, Zengo Kashinje, Jeremiah Seni.

**Visualization:** Dismas Matovelo, Richard Kiritta, Zengo Kashinje, Jeremiah Seni.

**Writing – original draft:** Colman Mayomba, Dismas Matovelo.

**Writing – review & editing:** Dismas Matovelo, Jeremiah Seni.

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
