## [Decision Letter · Decision Letter 0]

1 Mar 2024

PONE-D-24-00603Asymptomatic bacteriuria and its associated fetomaternal outcomes among pregnant women delivering at Bugando Medical Centre in Mwanza, TanzaniaPLOS ONE

Dear Dr. Matovelo,

Thank you for submitting your manuscript to PLOS ONE. After careful consideration, we feel that it has merit but does not fully meet PLOS ONE’s publication criteria as it currently stands. Therefore, we invite you to submit a revised version of the manuscript that addresses the points raised during the review process.

We look forward to receiving your revised manuscript.

Kind regards,

Seth Agyei Domfeh, PhD

Academic Editor

PLOS ONE

3. In this instance it seems there may be acceptable restrictions in place that prevent the public sharing of your minimal data. However, in line with our goal of ensuring long-term data availability to all interested researchers, PLOS’ Data Policy states that authors cannot be the sole named individuals responsible for ensuring data access (http://journals.plos.org/plosone/s/data-availability#loc-acceptable-data-sharing-methods).

Reviewers' comments:

Reviewer's Responses to Questions

**Comments to the Author**

1. Is the manuscript technically sound, and do the data support the conclusions?

Reviewer #1: Yes

Reviewer #2: Yes

2. Has the statistical analysis been performed appropriately and rigorously? 

Reviewer #1: Yes

Reviewer #2: Yes

3. Have the authors made all data underlying the findings in their manuscript fully available?

Reviewer #1: Yes

Reviewer #2: Yes

4. Is the manuscript presented in an intelligible fashion and written in standard English?

Reviewer #1: Yes

Reviewer #2: Yes

5. Review Comments to the Author

Reviewer #1: Reviewer report to the editor-in-chief of Manuscript titled: Asymptomatic bacteriuria and its associated fetomaternal outcomes among pregnant women delivering at Bugando Medical Centre in Mwanza, Tanzania

General Comment:

• The paper is well written grammatically correct…..

Comments/Questions for Clarification

Background

1. Reference 1, indicates prevalence of UTI among pregnant women globally is old, update to recent data.

Methods

1. Why women with multiple pregnancies are excluded? And what multiple pregnancies means?. Is to show multigravidity or twin, triple pregnancy at current pregnancy?

Results

1. Age is categorized as < 30 and ≥ 30 but, your nation indicates ‘ranging from 15 to 45 years’. Brief or correct.

2. Table 1, variable parity categorized as Multipara and Primigravida: are contradicting, gravidity and parity are different. Please correct or brief.

3. On table 2, 3 and 4, what are the empty boxes and * indicates in the multivariable logistic regression analysis column.

Reviewer #2: Overall Impression: The title is good and effectively conveys the focus of the research. The study's objective, which is to assess the prevalence of ASB and its correlation with unfavorable fetomaternal outcomes among pregnant women at Bugando Medical Center, is succinctly stated in the abstract. The study's findings hold significance for clinical practice and public health interventions aimed at improving maternal and neonatal health outcomes. The results section presents key findings regarding the prevalence of ASB among pregnant women, associated risk factors, and fetomaternal outcomes. The use of logistic regression analysis strengthens the study's statistical rigor, allowing for the identification of significant associations between ASB and adverse outcomes. Overall, the study provides valuable insights into the prevalence of ASB and its association with adverse fetomaternal outcomes among pregnant women. The study is technically sound and the data supports the conclusion. However authors should kindly take not of the following minor issues:

Minor Issues

1. References: the authors listed 39 references, 25 of which are over five years old. I suggest authors update their references with more recent studies that provide relevant information or findings related to their research topic. They can search for recent literature using academic databases, such as PubMed, Google Scholar, or specific journals in the field.

2. Minor grammatical errors: Line 74 -77: consider breaking it into two sentences for improved readability.

Line 88-89: consider revising "until" to "to ensure" for clarity.

line 118-122: consider changing "was" to "were" to agree with the plural subject "Antimicrobial susceptibility tests."

---

## [Author Response · Author response to Decision Letter 0]

3 Apr 2024

Responses to Reviewers comments have been attached/uploaded.

---

## [Decision Letter · Decision Letter 1]

1 May 2024

Asymptomatic bacteriuria and its associated fetomaternal outcomes among pregnant women delivering at Bugando Medical Centre in Mwanza, Tanzania

PONE-D-24-00603R1

Dear Dr. Matovelo,

We’re pleased to inform you that your manuscript has been judged scientifically suitable for publication and will be formally accepted for publication once it meets all outstanding technical requirements.

Kind regards,

Seth Agyei Domfeh, PhD

Academic Editor

PLOS ONE

Reviewers' comments:

Reviewer's Responses to Questions

**Comments to the Author**

1. If the authors have adequately addressed your comments raised in a previous round of review and you feel that this manuscript is now acceptable for publication, you may indicate that here to bypass the “Comments to the Author” section, enter your conflict of interest statement in the “Confidential to Editor” section, and submit your "Accept" recommendation.

Reviewer #1: All comments have been addressed

Reviewer #2: All comments have been addressed

2. Is the manuscript technically sound, and do the data support the conclusions?

Reviewer #1: Yes

Reviewer #2: Yes

3. Has the statistical analysis been performed appropriately and rigorously? 

Reviewer #1: Yes

Reviewer #2: Yes

4. Have the authors made all data underlying the findings in their manuscript fully available?

Reviewer #1: Yes

Reviewer #2: Yes

5. Is the manuscript presented in an intelligible fashion and written in standard English?

Reviewer #1: Yes

Reviewer #2: Yes

6. Review Comments to the Author

Reviewer #1: The study is well done, and it provides a timely information in the field, therefore, I suggest to be published as all the comments were addressed by the authors.

Reviewer #2: (No Response)

---

## [Editor Report · Acceptance letter]

10 Aug 2024

PONE-D-24-00603R1 

PLOS ONE

Dear Dr. Matovelo, 

I'm pleased to inform you that your manuscript has been deemed suitable for publication in PLOS ONE. Congratulations! Your manuscript is now being handed over to our production team.

Kind regards, 

on behalf of

Dr. Seth Agyei Domfeh 

Academic Editor

PLOS ONE